

# Effects of income and residential area on survival of patients with head and neck cancers following radiotherapy: working age individuals in Taiwan

Yu Cheng Lai[1,2,3,*], Pei Ling Tang[4,5,6,*], Chi Hsiang Chu[7] and Tsu Jen Kuo[2,8,9]

[1] Department of Orthopedics, Kaohsiung Veterans General Hospital, Kaohsiung, Taiwan
[2] Department of Marine Biotechnology and Resources, National Sun Yat-Sen University, Kaohsiung, Taiwan
[3] Department of Occupational Therapy, Shu Zen junior College of Medicine and Management, Kaohsiung, Taiwan
[4] Research Center of Medical Informatics, Kaohsiung Veterans General Hospital, Kaohsiung, Taiwan
[5] Department of Nursing, Meiho University, Pingtung, Taiwan
[6] College of Nursing, Kaohsiung Medical University, Kaohsiung, Taiwan
[7] Department of Clinical Trial Center, Kaohsiung Chang Gung Memorial Hospital, Kaohsiung, Taiwan
[8] Department of Dental Technology, Shu-Zen junior College of Medicine and Management, Kaohsiung, Taiwan
[9] Department of Stomatology, Kaohsiung Veterans General Hospital, Kaohsiung, Taiwan
[*] These authors contributed equally to this work.

Corresponding authors
Chi Hsiang Chu,
loveweib@gmail.com
Tsu Jen Kuo, tjkuo@vghks.gov.tw

## ABSTRACT

**Objectives.** The five-year survival rate of head and neck cancer (HNC) after radiotherapy (RT) varies widely from 35% to 89%. Many studies have addressed the effect of socioeconomic status and urban dwelling on the survival of HNC, but a limited number of studies have focused on the survival rate of HNC patients after RT.

**Materials and methods.** During the period of 2000–2013, 40,985 working age individuals (20 < age < 65 years) with HNC patients treated with RT were included in this study from a registry of patients with catastrophic illnesses maintained by the Taiwan National Health Insurance Research Database (NHIRD).

**Results.** The cumulative survival rate of HNC following RT in Taiwan was 53.2% (mean follow-up period, $3.75 \pm 3.31$ years). The combined effects of income and geographic effect on cumulative survival rates were as follows: high income group > medium income group > low income group and northern > central > southern > eastern Taiwan. Patients with moderate income levels had a 36.9% higher risk of mortality as compared with patients with high income levels (hazard ratio (HR) = 1.369; $p < 0.001$). Patients with low income levels had a 51.4% greater risk of mortality than patients with high income levels (HR = 1.514, $p < 0.001$).

**Conclusion.** In Taiwan, income and residential area significantly affected the survival rate of HNC patients receiving RT. The highest income level group had the best survival rate, regardless of the geographic area. The difference in survival between the low and high income groups was still pronounced in more deprived areas.

## INTRODUCTION

Globally, approximately 670,000 new diagnoses of head and neck cancer (HNC) and 350,000 HNC-related deaths are reported every year (*Ray-Chaudhuri, Shah & Porter, 2013*). HNC is the sixth most common cancer in Taiwan and the fourth most common among Taiwanese men (*Chang et al., 2017a*). Radiotherapy (RT) can effectively alleviate HNC (*Chu et al., 2016*; *Reeve et al., 2013*; *Wu et al., 2016*); however, adherence to RT is difficult for patients with severe toxicity associated with RT (*Thomas et al., 2017*), such as mucositis, taste disturbance, xerostomia, opportunistic infection, trismus, radiation caries, osteonecrosis of the jaw, and progressive periodontal destruction (*Cabrera, Yoo & Brizel, 2013*; *Kuo et al., 2016c*). These comorbidities impair chewing, swallowing, and speaking function.

The post-RT five-year survival rate of HNC patients varies widely—from 35% to 89% (*Hutcheson et al., 2014*; *Iyer et al., 2015*; *Langius et al., 2013*; *Lassig et al., 2012*); however, this large variation may result from differences in study designs and inclusion criteria. Many factors affect the survival of HNC patients after RT, including age (*Chang et al., 2013*; *Unal et al., 2015*), sex (*Olsen et al., 2015*; *Osazuwa-Peters et al., 2016*), race (*Osazuwa-Peters et al., 2016*), personal habits (e.g., smoking status, alcohol consumption, betel nut chewing) (*Chang et al., 2017a*), primary tumor site, tumor–node–metastasis stage (*Kreppel et al., 2016*), human papillomavirus status (*Chu et al., 2016*), therapy type (*Selzer et al., 2015*), nutritional status (*Chang et al., 2017a*), psychiatric disorders (*Unal et al., 2015*), urbanization (*Chang et al., 2013*), education (*Kjaer et al., 2013*), individual and neighborhood socioeconomic status (SES), and geographical area (*Chang et al., 2013*; *Chu et al., 2016*; *Kjaer et al., 2013*; *Wu et al., 2016*).

Many studies have found that a lower SES is associated with a lower survival rate among HNC patients (*Choi et al., 2016*; *Chu et al., 2016*; *Olsen et al., 2015*; *Osazuwa-Peters et al., 2016*; *Wu et al., 2016*). Other studies have revealed that neighborhood SES, geographical area, area-level socioeconomic position (SEP), and urban dwelling, all influence HNC patient survival (*Chu et al., 2011*; *Hagedoorn et al., 2016*; *Kuo et al., 2016a*; *Wong et al., 2017*). In general, lower neighborhood SES and rural residence are associated with lower survival rate among HNC patients. However, few studies have focused on post-RT survival rate (*Kuo et al., 2016a*).

Prediction of post-RT survival is fundamental for treatment planning by oral reconstruction dentists. Therefore, this study investigated the effects of SES (determined by income) and residential area on post-RT survival among working-age patients with HNC in Taiwan.

## MATERIALS & METHODS

### Data source and study cohort

Taiwan's National Health Insurance (NHI) program was established in 1995. With 23 million enrollees, it currently covers more than 99% of the Taiwan population. The data from Taiwan's NHI Research Database (NHIRD) are generally reliable and accurate (*Chang et al., 2017b*). We identified 66,626 patients with HNC who received RT during 2000–2013

from the registry of patients with catastrophic illnesses in the NHIRD. Of them, those with a prior history of cancer ($n = 3,131$), incomplete data ($n = 43$), RT procedure codes 36012B or 36011B <100 times in 75 days (since RT commenced; $n = 13,809$), age $\geq 65$ years ($n = 8,492$), and age $\leq 20$ years ($n = 166$) were excluded (*Kuo et al., 2016c*). Finally, 40,985 patients who received RT for HNC were included. The study was approved by Kaohsiung Veterans General Hospital (VGHKS15-EM10-02).

The applicable International Classification of Diseases, Ninth Revision, Clinical Modification (ICD-9-CM) codes specific to HNC were adopted (https://en.wikipedia. org/wiki/ Oral_cancer), namely lips (ICD-9-CM 140), tongue (ICD-9-CM 141), major salivary glands (ICD-9-CM 142), gums (ICD-9-CM 143), mouth (ICD-9-CM 144), other and unspecified parts of the mouth (ICD-9-CM 145), oropharynx (ICD-9-CM 146), nasopharynx (ICD-9-CM 147), hypopharynx (ICD-9-CM 148), other and ill-defined sites within the lip (ICD-9-CM 149), and larynx (ICD-9-CM 161). In addition, RT procedure codes (36012B or 36011B), specific for Taiwan, were included.

## Survival analysis

The start point for survival analysis was the index day, defined as the first day of RT, not the first day of diagnosis establishment. The endpoint of survival analysis was the day of death. For patients who survived until the end of the observation period, December 31, 2013 was considered the endpoint.

## Income and geographical area

The NHI premium depends on the income of the patients. Thus, although the NHIRD does not record patients' education level, it records their income. We used this to represent the income factor in our design. We categorized monthly income as follows: low, $\leq$US\$547 ($\leq$NT\$17,500); moderate, US\$547–781 (NT\$17,500–NT\$25,000); and high, $\geq$US\$781 ($\geq$NT\$25,001; the US\$"—NT\$ conversion is based on an average conversion rate of NT\$32 to US\$1 for 2015–2016). The geographical area was classified as Northern, Central, Southern, and Eastern (including the offshore island group) Taiwan (Fig. 1) (*Hung et al., 2015*).

## Other variables

Other variables included the date of RT administration (before or after January 1, 2009), tumor origin [oral (ICD-9-CM 140–145) or non-oral (ICD-9-CM 146-149,161)] (*Kuo et al., 2016b*), use in combination with conventional chemotherapy (cisplatin or 5-fluorouracil; yes or no), mandibulectomy or maxillectomy (yes or no), and excision of HNC malignant tumor within 3 months from the index day (yes or no). In patients with HNC of oral origin, the malignant neoplasm sites were the lips, tongue, major salivary glands, gums, mouth floor, and other unspecified parts of the mouth, whereas they were the oropharynx, nasopharynx, hypopharynx, unspecified pharynx, and larynx in patients with HNC of non-oral origin (*Kuo et al., 2016b*). Volumetric-modulated arc therapy (VMAT) was introduced in 2009 in Taiwan; the cutoff point in this study was also 2009.

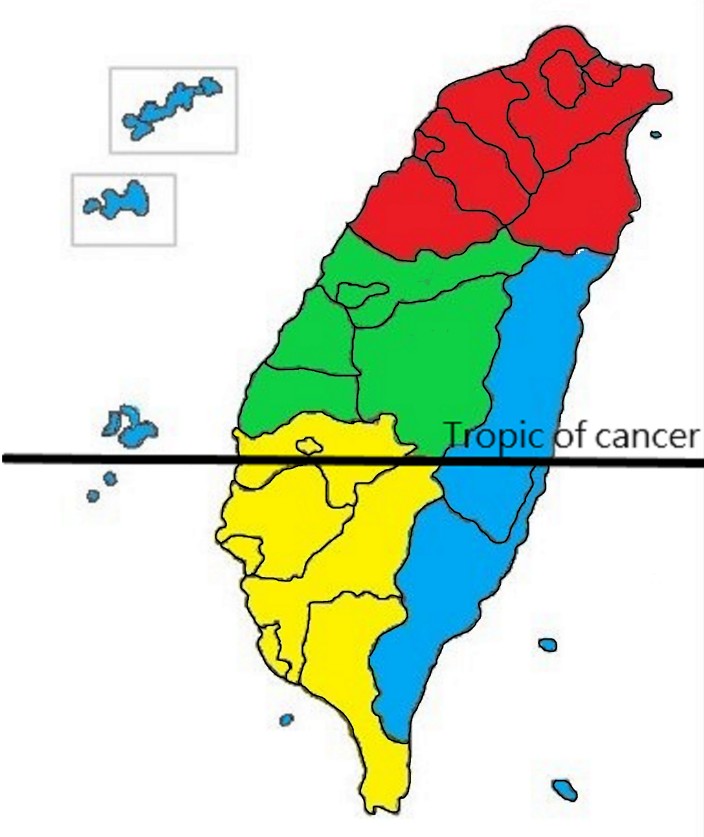

**Figure 1  Definition of residential area in Taiwan.** Red is designated for the northern area, green for the central, yellow for the southern, and blue for the eastern and offshore islands.

## Statistical analyses

All statistical analyses were performed on SPSS (version 20; SPSS Inc., Chicago, IL, USA). The Pearson chi-square test was used for analyzing categorical variables (sex, geographic region of residence, tumor origin, surgery, chemotherapy, and timing of RT), where as a one-way analysis of variance was employed for the continuous variable (age). The chi-square test of homogeneity was used for comparing survival rates until the end of the observation period between income levels and geographical areas. The $Z$-test with Bonferroni adjustment was used for post hoc comparisons between groups. The Kaplan–Meier method was used for survival analysis with variables limited to income levels and geographical areas only, whereas differences in the survival curves were identified using the log-rank test. A Cox regression model was adjusted for baseline covariates.

## RESULTS

### Demographic data and clinical characteristics

In total, 40,985 working-age HNC patients treated with RT (mean age, 49.23 ± 8.66 years; age range, 20.01–64.99 years) were included, and the overall survival rate was 53.2%

**Table 1  Baseline characteristics.**

| Variables | Income | | | | |
|---|---|---|---|---|---|
| | Low (*n* = 12,481) | Medium (*n* = 16,168) | High (*n* = 12,336) | Total (*n* = 40,985) | *p*-value |
| Mean age, yrs (±SD) | 48.45 (±9.01) | 49.89 (±8.46) | 49.17 (±8.50) | 49.23 (±8.66) | <0.001 |
| Residential area cases (%) | | | | | |
| Northern | 4,810 (38.5%) | 5,151 (31.9%) | 6,093(49.4%) | 16,054 | <0.001 |
| Central | 2,952 (23.7%) | 4,767 (29.5%) | 2,451(19.8%) | 10,170 | |
| Southern | 4,112 (32.9%) | 5,443 (33.6%) | 3,327 (27.0%) | 12,882 | |
| Eastern | 607 (4.9%) | 807 (5.0%) | 465(3.8%) | 1,879 | |
| Sex | | | | | 0.997 |
| Male (%) | M: 87.35% | M: 87.29% | M: 87.37% | M: 87.33% | |
| Female (%) | F:12.65% | F: 12.71% | F:12.63% | F:12.67% | |
| With tumor surgery (Around 3 months of index day) | 9.1% | 9.3% | 7.8% | 8.8% | <0.001 |
| With mandibulectomy or maxillectomy surgery (in 3 months before index day) | 5.02% | 5.40% | 3.74% | 4.78% | <0.001 |
| Timing of receiving R/T Before 2009 (%) | 55.04% | 57.42% | 57.50% | 56.72% | <0.001 |
| Origin: Oral cavity (%) | 44.85% | 45.58% | 36.74% | 42.70% | <0.001 |
| Combine chemotherapy(%) | 77.2% | 75.3% | 76.6% | 76.3% | 0.01 |

(mean follow-up period, 3.75 ± 3.31 years; Table 1) until December 31, 2013 (end of the observation period). The age range of the study cohort was limited to 20–65 years because the mandatory retirement age by law in Taiwan is 65 years. Low- and high-income HNC patients had a higher and lower proportion of tumors with oral origin, respectively (Table 1). As shown in Fig. 2, the effects of income and geographical area on the cumulative survival rates were in the following orders: high >moderate >low and Northern >Central >Southern >Eastern, respectively. Figures 3A–3C depicts Kaplan–Meier plots for overall survival, survival curves according to different geographical areas, and survival curves according to income levels, respectively. Figure 4 illustrates a Kaplan–Meier plot of survival of patients with HNC undergoing RT based on geographical area and income. Median survival in years was longest and shortest among patients residing in Northern and Eastern Taiwan, respectively (Northern >Central >Southern >Eastern; Table 2). Median survival was longer in the high-income group than it was in the low- and moderate-income groups. Significant differences were noted in the survival curves according to the geographical area (Table 3).

## Univariate survival analysis

As shown in Table 2 and Fig. 2, among the adult HNC patients (20 < aged < 65 years) residing in different geographical areas, survival was longer among the high-income group than among the low-income group ($p < 0.05$).

## Cox proportional hazard model

Results of the multivariate Cox proportional hazard model for the mortality of HNC patients receiving RT showed that residential area, income, sex, tumor origin, year of RT

| Geographic region | Income | | |
|---|---|---|---|
| | Low income | Medium income | High income |
| | Survival (%) | Survival (%) | Survival (%) |
| Northern | 54.2% | 55.9% | 65.3% |
| Central | 47.6% | 50.5% | 58.4% |
| Southern | 44.2 % | 47.6% | 56.2% |
| Eastern | 38.7% | 40.8% | 57.0% |
| All | 48.6% | 50.8% | 61.2% |

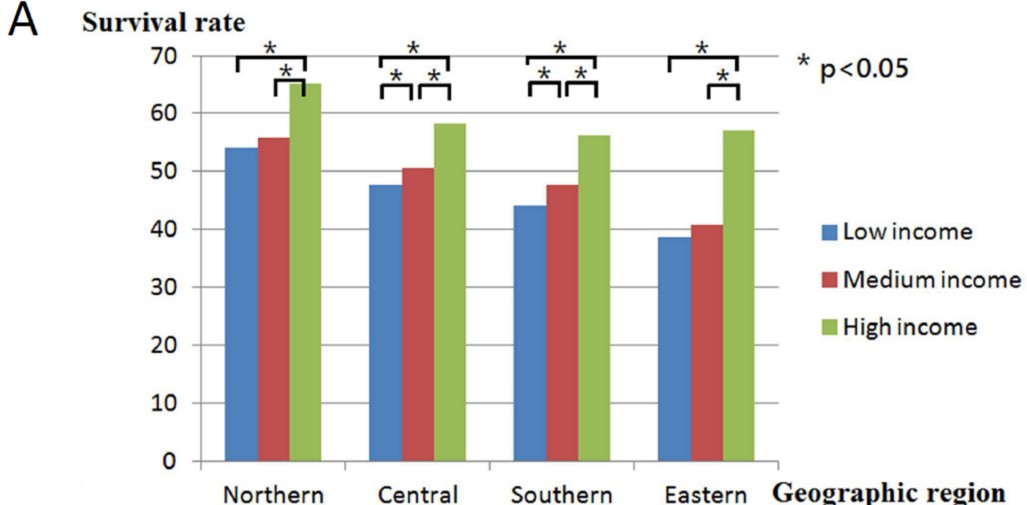

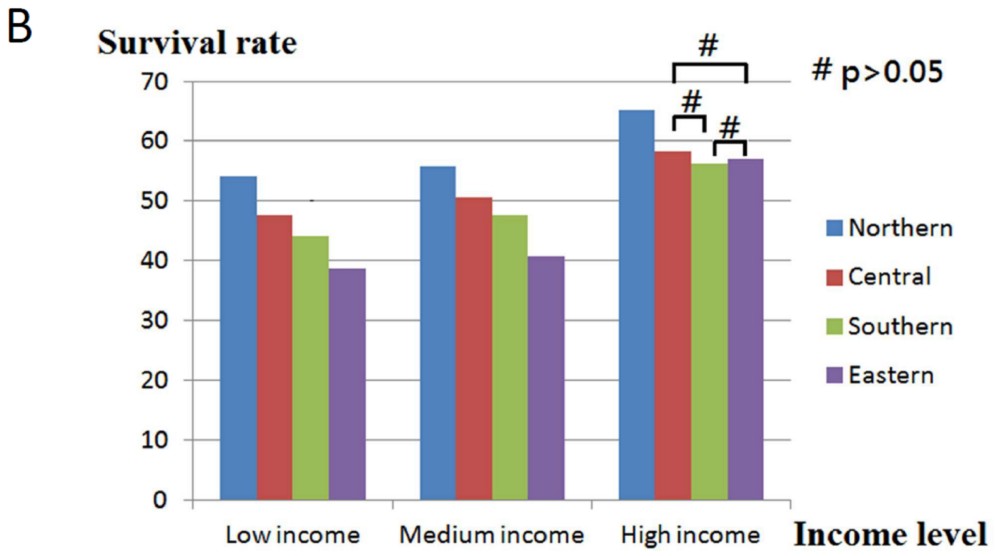

**Figure 2** **Description and comparisons of post-radiotherapy survival of head and neck cancer between geographic areas and income levels.** (A) Comparisons between low, medium and high income level in four geographic areas. (B) Comparisons between northern, central, southern and eastern areas in three income levels.

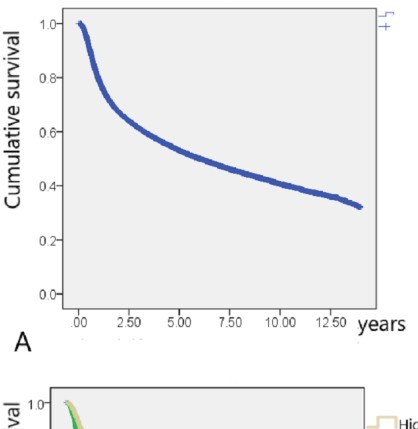

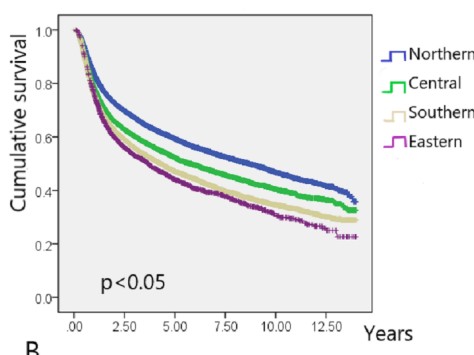

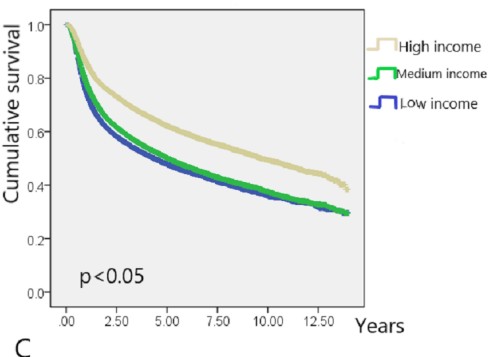

**Figure 3    Kaplan–Meier survival curve of HNC patients post radiotherapy.** (A) Kaplan–Meier survival curve of HNC patients post radiotherapy ($N = 40,985$). (B) Kaplan–Meier survival curves of HNC patients post radiotherapy for different residential area. (C) Kaplan–Meier survival curves of HNC patients post radiotherapy for different income level.

administration (before or after 2009), use of tumor excision surgery, and use of combined chemotherapy were associated with survival (Table 4).

Being male was most significantly associated with reduced post-RT survival of HNC patients [hazard ratio (HR) = 2.049, 95% confidence interval (CI) = 1.943–2.162, $p < 0.001$]. This was followed by oral origin (HR = 1.660, 95% CI [1.609–1.712]); lower income level (HR = 1.514, 95% CI [1.458–1.572]); conventional chemotherapy (HR = 1.504, 95% CI [1.452–1.558]); residential area in Eastern Taiwan (HR = 1.454, 95% CI [1.362–1.552]); timing of RT, mandibulectomy, or maxillectomy (HR = 1.215, 95% CI [1.137–1.299]); and no tumor excision surgery (HR = 1.181, 95% CI [1.120–1.246]; all $p < 0.001$; Table 4).

Patients with moderate income had a 36.9% higher risk of mortality than did those with high income (HR = 1.369; 95% CI [1.320–1.420], $p < 0.001$). Patients with low income had a 51.4% greater risk of mortality than did those with high income (HR = 1.514, 95% CI [1.458–1.572], $p < 0.001$). Patients residing in Central Taiwan had a 12.8% greater risk of mortality than did those residing in Northern Taiwan (HR = 1.128, 95% CI [1.087–1.171], $p < 0.001$). Patients residing in Southern Taiwan had a 40.2% greater risk of mortality than did those residing in Northern Taiwan (HR = 1.402, 95% CI [1.355–1.451], $p < 0.001$).

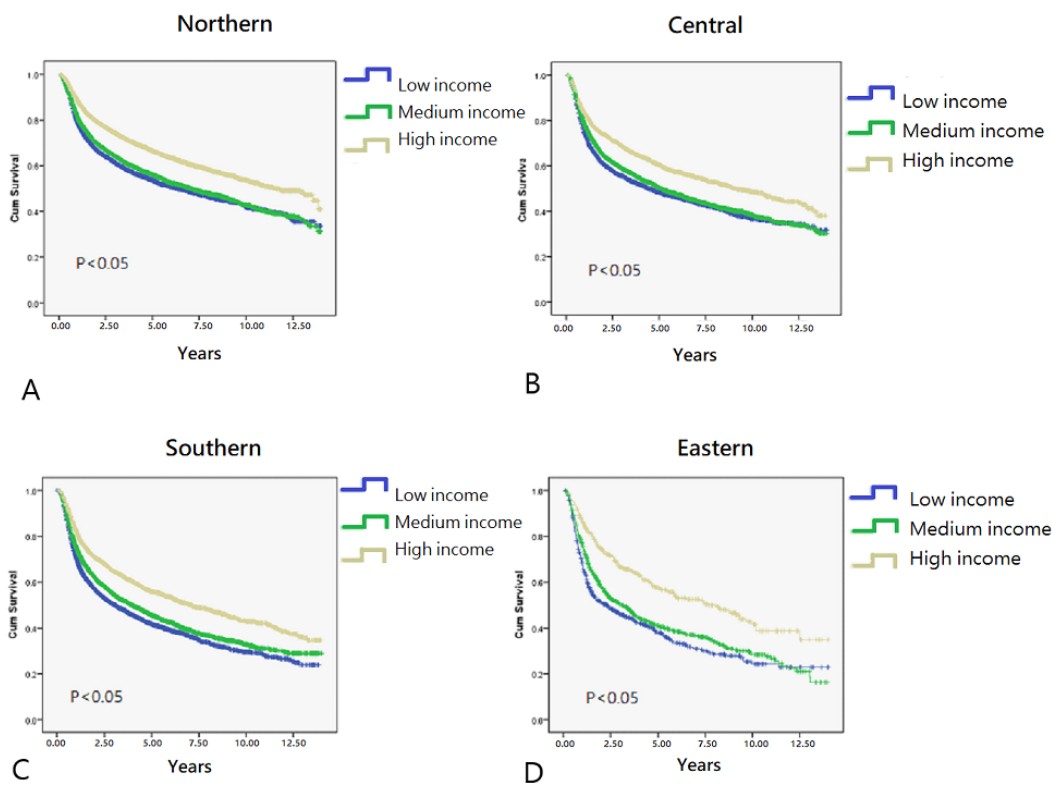

**Figure 4  Kaplan–Meier survival curves in different residential area.** (A) Northern area of Taiwan. (B) Central area of Taiwan. (C) Southern area of Taiwan. (D) Eastern area of Taiwan.

Patients residing in Eastern Taiwan had a 45.4% greater risk of mortality than did those residing in Northern Taiwan (HR = 1.454, 95% CI [1.362–1.552], $p < 0.001$).

Men had a 104.9% greater risk of mortality than did women (HR = 2.049, 95% CI [1.943–2.162], $p < 0.001$). Tumor with oral origins were associated with a 66.0% greater risk of mortality (HR = 1.660, 95% CI [1.609–1.712], $p < 0.001$) than were tumors with non-oral origins. The use of combined chemotherapy was associated with a 50.4% greater risk of mortality (HR = 1.504, 95% CI [1.452–1.558], $p < 0.001$) than was the use of no chemotherapy.

## DISCUSSION

According to the Surveillance, Epidemiology, and End Results database, the average diagnosis age of laryngeal, oral cavity, and pharyngeal cancer is 62 years. *Alvarenga Lde et al. (2008)* demonstrated that the average diagnosis age for HNC is 62 years in Brazil. However, we noted that the average diagnosis age of HNC in Taiwan is 51.84 years (*Kuo et al., 2016c*), much lower than that reported previously.

Individuals from the working-age group (20–65 years) provide the main source of family income and care; thus, any serious illness such as HNC can have a negative impact on their family, society, and country. The incidence of HNC is high in Taiwan. Most

**Table 2  Median survival years.**

| Residential area | Income level | Cases | Survival (years) | | |
|---|---|---|---|---|---|
| | | | Median | SD | 95% CI |
| Northern | | 16,054 | 8.680 | .218 | 8.252–9.108 |
| | Low | 4,810 | 6.360 | .341 | 5.691–7.029 |
| | Medium | 5,151 | 7.070 | .324 | 6.435–7.705 |
| | High | 6,093 | 11.610 | .436 | 10.756–12.464 |
| Central | | 10,170 | 5.740 | .199 | 5.351–6.129 |
| | Low | 2,952 | 4.450 | .311 | 3.841–5.509 |
| | Medium | 4,767 | 5.120 | .239 | 4.652–5.588 |
| | High | 2,451 | 9.140 | .576 | 8.011–10.269 |
| Southern | | 12,882 | 4.300 | .123 | 4.060–4.540 |
| | Low | 4,112 | 3.070 | .160 | 2.757–3.383 |
| | Medium | 5,443 | 3.950 | .168 | 3.621–4.279 |
| | High | 3,227 | 7.080 | .379 | 6.336–7.824 |
| Eastern | | 1,879 | 3.670 | .258 | 3.165–4.175 |
| | Low | 607 | 2.300 | .354 | 1.607–2.993 |
| | Medium | 807 | 3.080 | .297 | 2.498–3.662 |
| | High | 465 | 7.690 | .906 | 5.914–9.466 |
| | Total case | 40,985 | 6.030 | .101 | 5.831–6.229 |

**Table 3  Pair comparison of survival curve.**

| | Residential area | Income | Low | | Medium | | High | |
|---|---|---|---|---|---|---|---|---|
| | | | Chi-square | p value | Chi-square | p value | Chi-square | p value |
| Log Rank (Mantel-Cox) | Northern | Low | | | 5.123 | .024 | 202.379 | <0.01 |
| | | Medium | 5.123 | .024 | | | 146.615 | <0.01 |
| | | High | 202.379 | <0.01 | 146.615 | <0.01 | | |
| | Central | Low | | | 5.779 | .016 | 84.647 | <0.01 |
| | | Medium | 5.779 | .016 | | | 63.549 | <0.01 |
| | | High | 84.647 | <0.01 | 63.549 | <0.01 | | |
| | Southern | Low | | | 20.904 | <0.01 | 174.397 | <0.01 |
| | | Medium | 20.904 | <0.01 | | | 95.323 | <0.01 |
| | | High | 174.397 | <0.01 | 95.323 | <0.01 | | |
| | Eastern | Low | | | 4.380 | .036 | 54.975 | <0.01 |
| | | Medium | 4.380 | .036 | | | 36.183 | <0.01 |
| | | High | 54.975 | <0.01 | 36.183 | <0.01 | | |

HNC patients are men (91.3%) and aged 40–60 years (56.0%) (*Hsu et al., 2017*; *Hwang et al., 2015*). Taiwan has the highest oral cancer incidence worldwide. Among younger and male patients, oral and oropharyngeal cancers are more prevalent than hypopharyngeal and laryngeal cancers (*Hsu et al., 2017*). We focused on the survival of HNC patients who received a complete course of RT. Therefore, patients who received RT at a total dosage <60 Gy in 75 days were excluded according to our previous protocol (*Kuo et al., 2016c*).

**Table 4   Multivariate Cox proportional hazards model**

|  | Adjusted Hazard ratio | 95% CI | P |
|---|---|---|---|
| Random effect of income |  |  |  |
| High |  |  |  |
| Medium | 1.369 | 1.320–1.420 | <0.001 |
| Low | 1.514 | 1.458–1.572 | <0.001 |
| Residential area |  |  |  |
| Northern |  |  |  |
| Central | 1.128 | 1.087–1.171 | <0.001 |
| Southern | 1.402 | 1.355–1.451 | <0.001 |
| Eastern | 1.454 | 1.362–1.552 | <0.001 |
| Random effect of gender |  |  |  |
| Female |  |  |  |
| Male | 2.049 | 1.943–2.162 | <0.001 |
| Random effect of tumor origin |  |  |  |
| Origin:non-oral |  |  |  |
| Origin:oral | 1.660 | 1.609–1.712 | <0.001 |
| Random effect of tumor excision surgery |  |  |  |
| With surgery |  |  |  |
| Without surgery | 1.181 | 1.120–1.246 | <0.001 |
| Random effect of receiving R/T timing |  |  |  |
| After 2009 |  |  |  |
| Before 2009 | 1.219 | 1.180–1.259 | <0.001 |
| Random effect of mandibulec-tomy or maxillectomy surgery |  |  |  |
| With |  |  |  |
| Without | 1.215 | 1.137–1.299 | <0.001 |
| Random effect of chemotherapy |  |  |  |
| Without chemotherapy |  |  |  |
| With chemotherpay | 1.504 | 1.452–1.558 | <0.001 |

*Schwam, Husain & Judson (2015)* reported that the 3-year survival rate of HNC patients after adjuvant radiotherapy was 62.8%, higher than the survival rate of HNC patients who received a complete course of RT in the present study.

In general, HNC patients with lower incomes have lower survival rates than those with higher incomes (*Gupta et al., 2018*; *Subramanian & Chen, 2013*). Here, HNC patients with high income residing in Northern Taiwan had the highest overall survival rate, whereas those with low income residing in Eastern Taiwan had the lowest overall survival rate (Fig. 2). Income had a significant effect on the survival of HNC treated with RT, with the best survival rate being associated with the highest income, regardless of the area of residence. Both income and geographical area have been separately linked to the survival rate of HNC patients treated with RT (*Chu et al., 2016*; *Olsen et al., 2015*). According to data

published by the Taiwan government, life expectancy, concentration of medical facilities, and accessibility to medical resources are best in Northern Taiwan, followed by Central, Southern, and Eastern Taiwan (*Kuo et al., 2016a*). Because of worse transport infrastructure and a low density of medical resources, Eastern Taiwan is a medically deprived area. In the present study, regardless of residential area, income was significantly associated with median survival years and curves (Table 2, Fig. 4). Although the overall survival rate of patients residing in Eastern Taiwan was lower than that in other regions, the survival rate of the highest income group in Eastern Taiwan was even greater than that of the highest income group in Southern Taiwan (Fig. 2). However, no significant difference in the overall survival rate was noted among patients with the highest income in Eastern, Central, and Southern Taiwan (Fig. 2), probably because patients with higher income have a greater ability to cross regions and access better medical treatment and facilities (*Yi-Chen & Chin-Hung, 2010*). Our results also demonstrated that a higher income was associated with a higher survival rate in each regional area, and the differences in the survival curves and median survival years between the medium- and low-income groups were smaller than the differences between the high- and low-income groups or between the high- and moderate-income groups (Fig. 4, Table 2). We analyzed the interaction effect between income level and residential area, income, and surgery on post-RT mortality. Some interactions were discovered, and the trend was comparable to the original model—a more deprived residential area and lower income were both associated with higher post-RT mortality. Interaction effects between income level and surgical treatment were also noted. Among patients without tumor excision surgery, lower income was associated with higher mortality HR. However, in the high-income group, tumor excision surgery did not affect the post-RT mortality rate.

*Hagedoorn et al. (2016)* reported that among men aged 40–64 years with HNC in Belgium, survival was significantly lower for men with a low SEP and living in deprived areas. The differences in survival between the low- and high-SEP groups appeared less pronounced in more deprived municipalities (*Hagedoorn et al., 2016*). The main difference between our study and the study by *Hagedoorn et al. (2016)* is that we included both working-age men and women with HNC treated with RT. The difference in post-RT survival between low- and high-income groups was higher in more deprived areas in Taiwan, such as Eastern Taiwan.

Men exhibited 104.9% greater HNC-associated mortality than did women. Many studies have confirmed that survival is poorer among men with HNC than among women with HNC (*Choi et al., 2016*; *Chu et al., 2016*; *Olsen et al., 2015*; *Osazuwa-Peters et al., 2016*), which is consistent with the results of the present study. However, we discovered a much higher HR in men than that reported previously, which may have resulted from the following reasons: (1) women are more likely than men to seek medical care and comply with treatment regimens (*Osazuwa-Peters et al., 2016*), and (2) men are more likely to chew betel nut, which increases the risk of oral squamous cell carcinoma, an aggressive form of HNC (*Tung et al., 2013*; *Yang & Lin, 2017*). Approximately 10% of Taiwan's population habitually chews betel nut (~2 million people) (*Ko et al., 1992*). This percentage is higher in Southern and Eastern Taiwan, particularly among men (men: 16.5%; women: 2.9%), those

of lower SES, habitual smokers, alcoholics, and aborigines (*Chen et al., 2017*; *Chi-Pang et al., 2009*).

We noted that patients treated with either cisplatin or 5-fluorouracil chemotherapy had 50.4% greater risk of mortality than did who were not treated with chemotherapy. Cisplatin and 5-fluorouracil constitute standard chemotherapy for recurrent or metastatic squamous cell carcinoma of head and neck (SCCHN) (*Tahara et al., 2014*). Because more than 90% of HNCs in Taiwan are squamous cell carcinoma, we selected cisplatin and 5-fluorouracil as the chemotherapeutic variables. We assumed that most HNC patients received cisplatin and 5-fluorouracil to treat recurrent or metastatic SCCHN. Therefore, patients treated with chemotherapy had lower survival rate.

Intensity-modulated RT (IMRT) and VMAT provide superior target coverage, greater efficiency, fewer complications, shorter therapy duration, and less influence on the quality of life than do conventional RT and three-dimensional conformal RT (*Duarte et al., 2014*; *Lin et al., 2014*; *Tribius & Bergelt, 2011*). IMRT and VMAT have rapidly replaced conventional RT and three-dimensional conformal RT since 2009 in Taiwan (*Bedford & Warrington, 2009*; *OuYang et al., 2016*; *Zhang et al., 2015*). Therefore, the cutoff point in the present study was 2009.

### Limitations

Given that RT and CT for HNC are mostly outpatient treatments in Taiwan, the presence of dependents of working-age caregivers, such as children or parents, may have worsened treatment compliance. Although patients from deprived areas, such as Eastern Taiwan, often travel to other regions to receive medical services, the NHIRD only tracks the region of insurance application, which may be a patient's location of employment, rather than region of residence. Furthermore, the tumor–node–metastasis stage, nutritional status, education level, behaviors and habits, race, and faith of patients are unavailable in the NHIRD. The RT protocol type (conventional RT, three- dimensional conformal radiation therapy, IMRT, or VMAT), either palliative or curative, also affects the survival rate of HNC patients (*Marta et al., 2014*). We focused on the survival of HNC patients who received a complete RT course; however, the RT protocol was the NHIRD. Newer RT techniques, such as IMRT and VMAT, may not be simultaneously introduced in all four geographical areas of Taiwan. In a relatively deprived area such as Eastern Taiwan, the introduction of such techniques may well be delayed. This uncontrolled bias might confound the higher mortality discovered in Eastern Taiwan. Several studies have shown that being human papillomavirus positive is associated with better survival in patients with oropharyngeal squamous cell cancer (*D'Souza et al., 2016*; *Young et al., 2015*). These variables were not controlled or analyzed in the present study.

## CONCLUSION

Income and residential area significantly affected the survival rate of HNC patients receiving RT in Taiwan. The highest income group had the best survival rate, regardless of geographical area. The negative predictive factors for survival in HNC patients included

being male, tumor with oral origin, RT initiation before 2009, no tumor excision surgery, use of chemotherapy, and use of mandibulectomy or maxillectomy.

### Funding
This study was supported by grants (VGHKS104-133) from Kaohsiung Veterans General Hospital and the Taiwan Health Promotion Administration. The funders had no role in study design, data collection and analysis, decision to publish, or preparation of the manuscript.

### Grant Disclosures
The following grant information was disclosed by the authors:
Kaohsiung Veterans General Hospital.
Taiwan Health Promotion Administration.

### Competing Interests
The authors declare there are no competing interests.

### Author Contributions
- Yu Cheng Lai and Pei Ling Tang conceived and designed the experiments, performed the experiments, contributed reagents/materials/analysis tools, prepared figures and/or tables, authored or reviewed drafts of the paper, approved the final draft.
- Chi Hsiang Chu and Tsu Jen Kuo conceived and designed the experiments, performed the experiments, analyzed the data, contributed reagents/materials/analysis tools, prepared figures and/or tables, authored or reviewed drafts of the paper, approved the final draft.

### Human Ethics
The following information was supplied relating to ethical approvals (i.e., approving body and any reference numbers):

The Kaohsiung Veterans General Hospital granted ethical approval to carry out the study within its facilities (VGHKS15-EM10-02).

### Data Availability
The raw data are provided in the Supplemental Files.

### Supplemental Information
Supplemental information for this article can be found online at http://dx.doi.org/10.7717/peerj.5591#supplemental-information.

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
