# Peer review of "Effects of income and residential area on survival of patients with head and neck cancers following radiotherapy: working age individuals in Taiwan"

_PeerJ, doi:10.7717/peerj.5591_

## Round 0.1 · original submission · Major Revisions

Per my reading of the reviewers' comments, I feel that I agree with the strong criticism regarding the Discussion section of the manuscript, which is supposed to be the core part of the research. As of right now, the Discussion is not deep enough. The authors omitted important discussion of a variety of limitations their current study fails to address. Their treatment of data analyses also falls short of satisfaction. The use of English in the manuscript leaves a lot of room to improvement, and the general organization of the manuscript can be improved as well.

·

Basic reporting

The study is well written, adequately referenced, and the figures and tables appropriate.

I have a few minor suggestions related to technical details which I will put in the comments section below.

It was nice to include the raw data, but in SPSS format, it is unreadable to anyone who does not have SPSS (including this reviewer).

Experimental design

In the first sentence of the abstract it is unclear what the range 35-89% refers to (I assumed countries, but it turned out to be prior studies). Perhaps the number of studies considering survival after RT has been limited, but part of the reason may be that this is information that cancer registries collect (at least in the US). If I wanted to know five-year HNC survival after RT, I would not consult the literature; I would consult the SEER database, where I would get a definitive number that was population-based. If Taiwan lacks a cancer surveillance system where it is easy to find this information, then this paper serves a useful purpose in that regard.

Validity of the findings

The authors have identified significant independent effects of region and income on HNC survival in Taiwan, as well as effects related to treatment type. The data are robust and the statistical methods valid. The magnitude of the effects is rather large and would seem to lend itself to some potential intervention.

The authors suggest that the male-female differences may be related to betel nut use, but they do not say whether betel nut use varies by region or income level - it would seem to.

In the concluding paragraph, it would be useful not to simply restate the findings, but to address what, if anything, could or should be done to reduce the large disparities that were identified.

Additional comments

There is a strong effect of receiving chemotherapy having a higher mortality rate. This is counterintuitive as chemotherapy is supposed to be beneficial. The authors need to say something about this finding in the discussion.

Line 48-51: % higher risk of mortality and hazard ratios are redundant; either will suffice.

Line 71: The basis for the list of cancers being compared is unclear. There are other relatively common cancers for which HNC survival is better - why only list those for which it is worse? Maybe this sentence does not need to be here at all?

Line 72: see comment in box 2 above

Line 88: "However, only a limited number..."

Materials and Methods section: The definition of the four regions is never given. A map would be useful, but at a minimum there should be a citation.

line 98: It is unclear what exactly is meant by "unclear data".

line 99: These procedure codes are not from ICD9. A search for these codes yielded only other Taiwanese papers, so maybe this is a coding system specific to Taiwan? Please clarify.

Is 100 doses of radiation in 75 days considered the minimum standard?

line 107: If US dollar equivalents are given, the date on which this conversion was made should be listed, since it is variable - although it's not clear that it is even necessary to do this, since this is not a US-based journal nor is the readership US-based.

line 182-183: The reason for limiting the age to 65 should be moved to the methods. Same for the listing of which sites are considered part of oral cavity and which are not (lines 224-227) and the reason for the RT date cutoff of 2009 (line 240-241).

line 201: "access other hospitals"

Reviewer 2 ·

Basic reporting

Well-written throughout.
However, "survival" is at times 5-year (when other work is reported), and at times just "survival". It is not clear if it is 5-year survival that the authors are referring to, e.g.:
(lines 48-50): Patients with moderate income levels had a 36.9% higher risk of mortality as compared with patients with high income levels (hazard ratio (HR) = 1.369; p < 0.001). Patients with low income levels had a 51.4% greater risk of mortality than patients with high income levels (HR = 1.514, p < 0.001).
or whether it is "median survival" (e.g. lines 145 -- in which case the difference in survival should be reported in months or days, not %).
Similarly, this part is also confusing:
172 Males had a 104.9% greater risk of mortality than females (HR = 2.049, 95% CI
173 =1.943–2.162, p < 0.001).

It is not clear whether it was 5-year mortality or not.

Experimental design

Generally sound, national registry retrospective cohort design.
However, it is not reported whether Cox proportionate assumptions were met, and the lack of TNM staging, or even information whether the RT was with palliative or curative intent, is a serious limitation (i.e. we do not know if poorer patients are less compliant, have poorer nutrition that increases complications, or simply present late), a limitation which the authors themselves acknowledge.
Is the intent of RT (palliative vs curative) not present in the registry? If not, it should be stated. If it is, it should be in the model.

Validity of the findings

As mentioned above, the lack of TNM staging, or even information whether the RT was with palliative or curative intent, is a serious limitation (i.e. we do not know if poorer patients are less compliant, have poorer nutrition that increases complications, or simply present late), a limitation which the authors themselves acknowledge.

Reviewer 3 ·

Basic reporting

According to me this manuscript in its current form only partially passes the last bullet of this section (Self contained with relevant results to hypothesis). Specific suggestions are below:


1. Line 63: ‘as a result of this condition’ instead of ‘due to this disease’
2. Sentence beginning line 64 does not make complete sense. ‘Although’ part of the sentence should be followed by a counter part. do you just want to say that RT is associated with several complications, that impair quality of life? you can certainly structure this sentence much better and easier to read.
3. Line 65: ‘common’ word is not appropriate here. common in what context? in the general population? obviously No.. you could say frequently observed complications among patients receiving RT
4. in first paragraph, you should also give stats in the country of your study sample- since geography is one of your predictor variables
5. lines 74 to 81; it might be better if you reference each study next to the factor they studied or found to be associated with survival- it is hard to know go back to the study with all being listed together.
6. line 82, line 137: add “patients” after HNC- check throughout the paper
7. lines 81 to 88: are you repeating what you write in lines 74 to 81? why?
8. line 89: replace “determined” with examined
9. Restructure sentence- “therefore in the present study we examined the survival rate of HNC patients following RT among working age individuals in Taiwan, and assessed the influence of income and residential area on the association”
10. Please reword title of Figure 1- add patients, RT, in instead of with, area groups, comma to N
11. correct typos in title of figure 2- please check them all
12. line 180, 182- you should add this info about age and follow-up in results- typically not a part of Discussion section
13. line 181- correct to HNC patients
14. need to work on Discussion section – you do not add figure or table #s, CI, SDs in Discussion – they should be in results- only add those data from study that are necessary to make a point and keep the flow
15. Need to work a lot on Discussion section- besides language, you need to add literature in support and against your findings- how does your study fit in existing literature- why this study is relevant in Taiwanese population if earlier studies have shown similar results
16. It seems your Discussion is very much focused on sharing what you found- while you do that interpretation of your results in Discussion section, it has to be in done in the context of literature- what are the possible reasons for your findings, mechanisms, solutions, other factors- I don’t think you have to talk so much in length to describe what you found- it has to be an interpretation of results- eg lines 202 to 208- you mention all what you found in different ways- this should be in results- Discussion section is meant to provide an interpretation and what did you add to knowledge, acknowledging the limitations and future research agendas.
17. lines 217-220- again all this data should not be in this section
18. line 217- reframe- correct grammar
19. line 209- among male HNC patients- use appropriate language
20. lines 198 to 202- reference? need to restructure, correct grammar
21. lines 221-223- should be in results
22. lines 223- 228- should be in methods- discussion is not where you tell this
23. lines 228- talk in flow- you mentioned about males and females in lines 213 -215 above
24. lines 231- 236- it seems that you wanted to tell factors why HR was higher in your study, but the factors you say there are not the ones who looked into. are you simply stating the reasons why survival might be lower in males than females? if so, please clarify the language and words you use

Experimental design

According to me this manuscript in its current form only passes the first, partially second and third bullets of this section. Specific suggestions are below (some of them might be in Basic Reporting section above):


1. Line 97: explain more about the dataset- how do patients make into this database? collected from hospitals? voluntary reporting or mandated? repeat visits? how you accessed data? how are ICD9 codes assigned? is this administrative dataset for billing? how are income and geographic are and other variables recorded- self reported ..etc
2. Line 98: what do you mean by “unclear data”?
3. why did you exclude the ages you did? reference for definition of working class age groups? you could structure this better- why write >=20 years in line 95 and >=65 years in line 98?
4. usually you would begin by explaining the data set – what sample size you began with and then what were your exclusions (and why)- and how many you excluded for each reason- so readers know exactly the flow of how you zeroed down to the final analytic sample from your 66,626 patients. See flow-charts for trials- something on those lines should be used. your sample selection is unclear as of now.
5. line 99: explain what are these procedure codes, what do you mean by <100 times? what is the rationale and why you are excluding them
6. line 100- follow a pattern of writing numbers – 40,985- add comma; also line 133; check throughout the paper
7. line 103: explain ICD in full form
8. lines 103 to 105-include in your flow chart- how many had code for HNC and from that how many received RT and then your exclusions one by one
9. line 104- I think you should explain the cancers you included and not just the codes.. just say we included patients with cancer of tip of tongue (ICD9 code xx), floor of mouth (ICD9 code yyy)………. etc.. it is good for reproducibility of study
10. also for flow of the paper- these codes should come earlier in the beginning of flow chart- definitions in terms of codes of HNC, RT

11. line 107: “categorized” instead of ‘defined”
a. add colon after name of each category- it is confusing to read otherwise
i. eg- low income: USD 547 per month; moderate income: uSD 547-781 per month…………. etc
12. line 113- what is the rationale behind 2009 for date of RT administration
13. line 123- for consistence add the continuous variables in parenthesis

14. you don’t describe the start and end periods of follow-up for survival analysis- they should be made more explicit in methods section

15. line 124: you should state how the underlying assumptions for the test are met- if you checked them or not to determine appropriateness of test
16. line 129: what were the adjusting variables- indicate here for clarity

17. Figure 1- what test did you use? I am guessing it is chi square test within the specific population groups- eg. among all in northern region, tabulating survival and income- so you would get 1 p value for that statistic- I don’t get how you have small brackets for specific income groups? Eg. p value for distribution of survival across income levels among people in Northern region was <0.05, but what does the small arrow on medium and high income there refers to?
18. Figure 1- second graph- did you mean p<0.05 or >0.05 for #

19. line 140- figure 1 does not give the overall survival in each group- it breaks up data by income and geographic region- if you want to say ‘as shown in fig 1’- add overall survival rate in each group in table maybe- row and column total

20. line 152- why do you say <65 years again- you have defined your sample population- saying some of them again makes it confusing- for eg here it seems that you have included all those who are <65 years- did you include those <20 years for this analysis?
21. need to correct sentence framing
22. line 156- it is not logistic regression
23. line 174, 176- compared to what? always mention comparison group even though it is in tables

24. lines 232- what do you mean by greater significance? Significance is either yes or no based on your p value- did you mean to say higher HR?
25. line 240- I think this should go in methods- why you selected that cutoff- also dis VMAT completely replace other modalities after 2009, so you are confident that those who received after 2009 will have received VMAT?- if not, there is a bias and assumption which should either be addressed (if you can identify what treatment was given) or acknowledged as a limitation

Validity of the findings

According to me this manuscript in its current form only partially passes second bullet (Data is not adequately controlled) and third bullet. Specific suggestions are below:


1. Racial disparities in survival of HNC patients are well known- did you account for that?
2. Age group should also be adjusted for
3. line 180- you never mentioned about universal insurance system before this point- should explain this in Methods- if it is universal insurance why do you see income based differences? Did you look at other SES factors- education, occupation etc?
4. line 183- I think you included those <65 years, did you include those 65 years as well- be specific about exclusions – was it <65 years or <=65 years?
5. line 210-211- how did they define “deprived areas”. you say in live 216 that you found associations in deprived areas- what was your definition?
6. lines 209 to 216- but you did not stratify by gender- if you are stating this paper it seems that gender specific associations might have been helpful.
7. VMAT introduction date makes me question the validity- are we sure that all patients who received treatment after 2009 got VMAT? if not there is a significant bias there.
8. also it is possible that “deprived” areas like eastern were slower to get VMAT- it was not that common in those areas as compared to Northern- and that is why risk of mortality is higher in those regions. how do you account for that? I do not think stratifying by 2009 date of treatment will take care of that noise if it was not a complete replacement by VMAT after 2009. There is an important confounding in that case.
9. you need to elaborate your limitations – there are several other limitations and assumptions as mentioned above in my comments, and they should be acknowledged here – they can significantly affect the results and their interpretations
10. if you do not have data on HPV, at least talk about it in Discussion or limitations , as HPV is becoming a significant risk factor for HNC especially oro-pharyngeal cancers and HPV positive tumors are directly related to prognosis- they have better survival

Additional comments

1. OVERALL- there is a lot of scope to improve language all over the paper- regarding selection of words and connectors. I have also made some suggestions, but authors should read through the paper carefully to improve the language- correct gramma and typos
2. Make sentences consistent- comma for numbers, etc.- have mentioned examples in comments above
3. figures do not seem to be of adequate resolution

---

## Round 0.2 · Minor Revisions

Please revise your manuscript based on the reviewers' suggestions. Please make sure you mark your revision for comparison purposes.

·

Basic reporting

This is a second review of a paper I previously reviewed. I am placing all of my comments in the general comments box below.

Experimental design

This is a second review of a paper I previously reviewed. I am placing all of my comments in the general comments box below.

Validity of the findings

This is a second review of a paper I previously reviewed. I am placing all of my comments in the general comments box below.

Additional comments

The authors have adequately responded to each of my comments. (I was unaware that SEER*Stat stopped including radiation data in its public-use database in 2016; they do offer it in a supplemental database, but that detail is not important for this paper).

While the quality of the English in the original submission was adequate, with the need for minor corrections noted by the reviewers, the quality of the new text (in red) is more of a problem. Sentences are poorly constructed and do not always follow logically from one to the other and there are many typos. For example, line 189 "according", line 225 "medical", line 251 "squamous". This requires attention before publication.

Reviewer 2 ·

Basic reporting

Generally good writing, with some typos, e.g.:
"although this large deviation may result from" (suggest "variation" rather than "deviation")
"differences in study design" (remove the "s" in designs)
"Accorind to " ("According to")
"It's generally agreed that HNC pateints "
"This may result from that patients with higher income level "
Also, since the patients were not enrolled in the study, but registry data was used, suggest not to use the word enrolled.

Experimental design

Generally much improved. However, suggest the following:
1. Time from RT, rather than time from diagnosis, is used; there is nothing to tell us if
a. the time was not available (in which case, it should be mentioned as a limitation);
b. the time was available but not examined (for what reason);

2. There should be an attempt (e.g. sensitivity analysis) to explore the following interactions statistically:
a. socioeconomic level and geographic region
b. socioeconomic level and other modalities of treatment (chemotherapy, surgery), since the data is available

and whether these change the findings should be reported

Validity of the findings

Generally valid for the specific population studied.

Additional comments

Suggest highlight in the discussion that radiotherapy and chemotherapy, unlike surgery, require a protracted period of compliance throughout the outpatient treatment (please confirm that in Taiwan, radiotherapy and chemotherapy are indeed outpatient treatments where the patient stays at home), and the presence of young or invalid dependents (children or patients'' parents) and working-age caregivers (spouses) may worsen the compliance.

Reviewer 3 ·

Basic reporting

There are multiple typos and grammatical errors in the correct text.
I still think Limitations section needs more insight- please see attached for details.

Experimental design

please see attached for details

Validity of the findings

please see attached for details

Annotated reviews are not available for download in order to protect the identity of reviewers who chose to remain anonymous.

---

## Round 0.3 · accepted · Accept

In my review of the authors' response to the second round of review, I believe the authors have addressed the comments adequately. I do notice, however, a few language issues throughout the manuscript, and would suggest the authors seeking professional language editors for further polishing.